# Application Progress of Modified Chitosan and Its Composite Biomaterials for Bone Tissue Engineering

**DOI:** 10.3390/ijms23126574

**Published:** 2022-06-12

**Authors:** Yuemeng Zhu, Yidi Zhang, Yanmin Zhou

**Affiliations:** Department of Oral Implantology, School of Dentistry, Jilin University, Changchun 130021, China; yuemeng21@mails.jlu.edu.cn

**Keywords:** modified chitosan, cross-linking, structure modification, bone tissue engineering

## Abstract

In recent years, bone tissue engineering (BTE), as a multidisciplinary field, has shown considerable promise in replacing traditional treatment modalities (i.e., autografts, allografts, and xenografts). Since bone is such a complex and dynamic structure, the construction of bone tissue composite materials has become an attractive strategy to guide bone growth and regeneration. Chitosan and its derivatives have been promising vehicles for BTE owing to their unique physical and chemical properties. With intrinsic physicochemical characteristics and closeness to the extracellular matrix of bones, chitosan-based composite scaffolds have been proved to be a promising candidate for providing successful bone regeneration and defect repair capacity. Advances in chitosan-based scaffolds for BTE have produced efficient and efficacious bio-properties via material structural design and different modifications. Efforts have been put into the modification of chitosan to overcome its limitations, including insolubility in water, faster depolymerization in the body, and blood incompatibility. Herein, we discuss the various modification methods of chitosan that expand its fields of application, which would pave the way for future applied research in biomedical innovation and regenerative medicine.

## 1. Introduction

Human bone is a three-dimensional (3D) composite porous structure with several roles within the body consisting of 30% organic matter and 70% inorganic matter [1]. It performs the functions of protection, locomotion, storage depot for calcium and phosphate in the body, housing for bone marrow, and structural integrity to the body [2,3]. However, with aging, accidents, and bone illnesses, bone deformities’ high frequency and handicaps have become prevalent difficulties in clinical orthopedics and a serious societal issue. The demand for bone grafting techniques has constantly been increasing all across the globe [4]. Critical-sized bone defects are believed to cause an unfavorable wound environment and therefore cannot undergo spontaneous healing [5]. Conventional tactics such as autografts or allografts, alone and in combination, are still used for bone repair and regeneration even though only limited therapeutic efficacy can be achieved in clinical settings [6]. However, the applications of autografts are limited by the donor site morbidity, insufficiency, and low availability of bone grafts [7], and allografts hold the risk of disease transmission and immunoreaction [8].

Recently, advances in tissue engineering and technologies in this field could provide more efficient treatment [9,10]. Bone tissue engineering (BTE) has arisen as an alternative and attractive approach to overcoming the disadvantages of conventional bone grafts for treating critical size defects through scaffolds, seed cells, and biologically active molecules. It offers prospective alternatives to autografts and allografts by making effective use of the regulation of tissue regeneration. BTE aims to accommodate natural and synthetic biomaterials to design regenerative bone scaffolds encapsulated with specific essential components. The scaffolds should maintain the following properties for adequate bone tissue regeneration, including biocompatibility, suitable mechanical properties, pore interconnectivity, and bioresorbability [10]. The ideal biomaterials for bone scaffolds render essential physical and chemical properties for tissue regeneration compatibilities, such as a large surface area, suitable mechanical strength, stability, and the improvement of cell adhesion, proliferation, and differentiation [11]. So far, several materials with biological properties have been developed as BTE scaffolds materials, encompassing natural polymers, synthetic polymers, bioceramics, biodegradable metals, and carbon-based nanomaterials [12,13,14].

The marine environment accounts for about half of the global biodiversity, and nearly 70% of the earth’s surface is covered by oceans [15]. Among the many polysaccharides available from the marine environment, chitin stands out for its availability as it is the second most abundant natural polymer after cellulose [16]. Although chitin is abundant and possesses special functional characteristics such as biocompatibility, bioactivity, and biodegradability [17,18,19,20], its use is limited due to its poor solubility and mechanical strength. This makes chitin not very serviceable and shifts attention towards chitosan (CS) [21]. As a positively charged low-cost natural polymer [22], CS is the main derivative of chitin. Shrimp and crabs are the most common sources cited in the literature as the raw material for chitosan preparation. Existing studies regard the by-products of crustacea such as the lobster cephalothorax as a suitable source for chitosan preparation on an industrial scale [23,24]. CS consists of 2-amino-2-deoxy-β-d-glucopyranose and N-acetylglucosamine units and displays extraordinary properties. The amine groups of CS will boost cell adhesion and growth. CS-based biomaterials have shown promise when applied to tissue engineering and regenerative medicine because of their excellent biocompatibility [25], antibacterial properties [26], and osteoinductive ability. Owing to the vast similarities with glycosaminoglycans (GAGs), CS can increase the bone regeneration rate. One of the most vital manifestations of the biocompatibility of CS scaffolds is that they can promote cell adhesion, proliferation, and differentiation [27,28]. They can maintain a cell’s normal activities and promote tissue regeneration. Meanwhile, the portable amino groups of CS bind with the negatively charged mucin (cell membrane), resulting in mucoadhesion [29]. CS can additionally improve the osseointegration and corrosion resistance of metal implants, which is indispensable for the long-term survival of internal implants in vivo [30]. However, it is not likely that CS alone can be used to make up the scaffold structure because of the non-affinity of water [31]. It also does not fully meet the mechanical requirements of the implant site. Studies have shown that functional groups of the polysaccharide backbone serve as anchoring sites for chemical modifications, generating versatile scaffolds of great significance in the biomedical field [32]. The possible modification of the major functional groups (OH and NH2) at carbon-2 and carbon-6 permits the preparation of various CS derivatives of improved chemical and physical characteristics for specific uses and functions [33]. CS has high-level osteoconductivity but low-level osteoinductive activity [34]. Therefore, it would be optimal to modify CS or from compounds with other biomaterials to enhance the solubility, mechanical properties, and antibacterial properties of the composite scaffold for BTE.

To date, there have been considerable efforts to explore modified CS-based materials for BTE. Table 1 summarizes the recent studies and essential results of modified CS-based materials used in BTE to enhance the efficacy of bone repair and regeneration. There are several key considerations in designing a modified CS-based system to achieve efficient and efficacious therapy for future clinic applications. Therefore, in this review, we report the current advances and applications of modified CS and its composite biomaterials for BTE. This review is divided into six parts: (1) anti-inflammatory effect and safety of CS; (2) physically cross-linked CS for BTE; (3) chemically cross-linked CS for BTE; (4) enzymatically cross-linked CS for BTE; (5) structure modification of CS for BTE, and (6) CS grafted with biodegradable polymers for BTE. It is believed that this review article could help researchers understand the whole picture of progress, recent advances, and future prospects with modified CS-based scaffolds for bone defect treatment.

## 2. Post-Implantation Complications and the Safety of Chitosan

Despite the phenomenal success of implants of BTE particularly in the realms of dentistry and orthopaedics, there are still challenges to overcome. The failure of implants resulting from infection, prosthetic loosening, and non-union continue to be the most serious examples [35]. As we know, new biomaterials for bone regeneration may induce a foreign body reaction (FBR) around the implant itself and may cause systemic inflammation [36]. After biomaterial implantation, several factors have a significant impact on bone tissue repair, inter alia macrophage–osteoblast cross-talk, soluble environmental factors, and surface properties of the implant [37]. The success of biomaterial implantation depends highly on the implant’s macrophage polarization [38]. Macrophages are known to be involved in the in vivo biodegradation of resorbable polymers through the release of reactive oxygen intermediates, enzymes, and acids [39]. They also exert an immunomodulatory effect on osteogenic differentiation, inducing bone formation [40,41]. Macrophages secrete an incredible amount of signaling molecules that initiate an inflammatory response against the foreign body and regulate cell migration and differentiation, tissue remodeling, and new blood vessel formation [42]. The pro-inflammatory mediators, TNF-α and IL-1, may lead to excessive bone resorption by the priming of osteoclasts [35,43,44]. The local damage caused by implantation and the presence of a foreign material elicits an immune response that uses chemokines to attract circulating monocytes to the area, which differentiate into activated macrophages that release TNF-α, IL-1, IL-6, and M-CSF [45]. Under chronic inflammation, these macrophages exist in abundance and have been found to differentiate further into pre-osteoclasts and then into mature osteoclasts in the presence of M-CSF. Furthermore, these macrophages can express chemokines to enable self-activated osteoclastogenesis. It is worth noting that macrophages isolated from peri-prosthetic tissue have been found to differentiate into mature osteoclasts without the company of MSCs or osteoblastic cells [46].

Bone injury and chain reactions mediated by free radical species (reactive oxygen species, ROS) generation are other critical factors. ROS affect the long-term stability of bone/implants and mediate the apoptosis of osteoblasts and osteocytes, leading to osteoclastogenesis and thereby favoring bone resorption [47]. Post-implantation, vicinity acquired oxidative stress and bacterial infections lead to apoptosis with eventual bone-resorption and implant failure, respectively. After the implantation, in such patients (disease, fracture, and age being the contributors), the oxidative stress secludes the material from the surrounding tissue and also leads to cytotoxicity [47]. An in vivo study on rabbits reported that significant levels of oxidative stress are induced in the tissues surrounding a bone implant (especially, ceramic and titanium in comparison to polyethylene) [48].

Several studies have been developed in order to combat the aforementioned complications. CS and its derivatives show an intensified anti-inflammatory cytokine induction [49]. Studies have demonstrated that CS-based scaffolds bioactivated with osteoinductive signals can inhibit in vitro inflammatory responses [50]. The scaffolds show anti-inflammatory activity also in in vitro co-cultures, which better mimic the in vivo damaged bone microenvironment. CS-based bioactivated scaffolds may inhibit the synthesis of inflammatory mediators such as IL-1β [51], reduce oxidative stress metabolites, decrease pro-inflammatory cytokine (TGF-β) levels, and promote anti-inflammatory marker generation (IL-10) in hMSCs. CS with a low molecular weight could favor macrophage polarization to the M2 phenotype and its bioactivity in the NF-κB and FGF-2 pathways [52].

Several preclinical and clinical experiences further confirm the biological properties and safety in practice for tissue engineering. Ueno H et al. evaluated the effect of CS as an accelerator of wound healing, and experimental open skin wounds on the dorsal side in normal beagles were made [53]. The reports verified that CS could activate immunocytes and inflammatory cells such as PMN, macrophage, fibroblasts, and angioendothelial cells. A randomized controlled trial [54] evaluated the effect of CS combined with *Dysphania ambrosioides* (A) extract on the bone repair process in vivo. Results showed that when CS-based spheres exhibited the ability to guide bone repair and osteoinduction they stimulated osteogenesis by recruiting osteoprogenitor cells [55]. Feng Liu et al. investigate the role of CMCS in knee arthroplasty [56]. Data confirmed that CMCS could effectively inhibit the inflammatory response around the prosthesis and osteoclast activation and promote osteogenesis by interfering with the osteoprotegerin and the receptor activator of nuclear factor kappa-Β ligand or the receptor activator of the nuclear factor kappa-Β signaling pathway.

In the investigations of Chih-Hsin Wang et al. [57], it was revealed that CS dressing had superior procoagulant and antimicrobial properties to regular gauze-type surgical dressing in patients with surgical wounds. For BTE, CS has been revealed to stimulate a rapid osteoblast response, displaying rapid cell spreading and cytoskeleton reorganization through the clinical trial developed by Antonia G Moutzouri [58].

These preclinical and clinical trials demonstrate potential for using this biomaterial for bone tissue regeneration purposes and open perspectives for further research to determine the mechanisms of these interactions and to develop bioactive scaffolds for BTE. The overviews of the trials above are displayed in Table 2.

## 3. Fabrication Strategies

Recently, manufactured scaffolds for BTE have been prepared by various methods such as electrospinning (ES) [59], self-assembly [60], and 3D bioprinting [61,62]. ES has been used for decades to generate nano-fibers via an electrically charged jet of polymer solution [63]. It is a process that utilizes an electric field to control the deposition of polymer fibers onto target substrates. The ES techniques include the sacrificial components method, wet-electrospinning method [64], cryogenic electrospinning method [65], dispersion-shaping method [66], gas-foaming method [67], and electrohydrodynamic printing method [68]. These advanced ES techniques can offer a 3D microenvironment to facilitate cell colonization inside scaffolds, enhance nanofibers’ mechanical strength, mimic the extracellular matrix (ECM) [69,70], and support nutrient and waste exchanges [71]. Amongst available biopolymers, CS and its naturally derived composites have been widely adapted for TE applications. Evidence supports the favorable properties and biocompatibility of CS–ES composite biomaterials for BTE [72,73]. Compared with pure CS membranes, CS nanofibers prepared by electrospinning have exhibited an excellent affinity for osteoblasts, which can facilitate osteoblast proliferation and maturation as well as upregulate osteogenic gene expression [74,75,76]. However, translating the chitin/chitosan nanofibers from laboratory to clinical application needs further research.

Along with the rapid development of organizational engineering, 3D printing technology, an additive fabrication method, is considered to be a new paradigm of bone tissue engineering and biomanufacturing [77,78]. 3D printing is a process for constructing 3D physical objects from digital models mimicking a natural-like extracellular matrix through the successive layer-by-layer deposition of materials [66,79]. The technique involves the deposition of a mixture of living cells and biomaterials (e.g., hydrogels) [80]. These printing approaches include FDM [81], stereolithography [82], different nozzle extrusion-based 3D printing technologies [83,84,85], and low-temperature manufacturing [86,87]. To date, several biomaterials, including bone and cartilage tissues, cardiac tissues and heart valves, neural, lung, liver, pancreatic, skin, retinal, vascular, and composites tissues have been fabricated through this technology [88]. Finely printed scaffolds can mimic the macro-and micro-structures of bone. The mechanical properties of the scaffolds could be regulated by structure designing, and the biological activity and degradation of the scaffolds can be adjusted through chemical composition [89]. Owing to the aqueous solubility of CS in an acidic environment, it is largely utilized for bioprinting applications, where it accounts for 4% of total polymer distribution used for bioink preparation [90]. The CS chains expand into a semi-rigid rod confirmation due to ionic repulsion between the charged groups (NH3+), and thus the CS ink exhibits shear thinning behavior under low shear rates at 25 °C, leading to better flow through the needle [91], which is beneficial for the extrusion-based 3D bioprinting [92]. CS-based hydrogels hold great promise for the development of 3D bioprinting inks to fabricate engineered constructs [93]. The chemical modification of CS through its high number of amino and hydroxyl groups improves its water solubility and facilitates formulation development. Neutralization and gelation under alkaline conditions, derivatization, cross-linking, or a combination with other polymers, are required for CS to be printed. For instance, hydroxyapatite (HAp) [94], pectin [95], β-TCP [96], graphene oxide (GO) [97], and various cross-linkers have been incorporated into CS for mechanical reinforcement of CS-based inks or 3D printed scaffolds in bone tissue engineering.

However, there are still a lot of challenges that need to be overcome before the 3D printing technology can be adopted as a common fabrication technology and can achieve its full potential. The limited variety of available, environmentally friendly, and printer-friendly materials is a key barrier to the wide-scale adoption of 3D printing technologies.

## 4. Modification Methods of Chitosan for BTE

There are numerous protocols for the preparation of modified and hybrid CS scaffolds, which attract huge interest, particularly in BTE [98,99,100]. Among different methods of modified CS, cross-linking, structure modifications, and grafts with biodegradable polymer are the basic approaches to producing BTE scaffolds. Depending on the nature of the polymeric backbone and the functional groups, CS could be cross-linked by using various methods, including physical, chemical, and enzymatic approaches and a combination of these. The primary purpose of cross-linking is to facilitate the biomechanical properties of scaffolds by the formation of a firm network in the polymeric matrix [101]. Furthermore, cross-linking can also modify the antigenic sites of natural materials and reduce their antigenicity [102]. Physical cross-linking pathways take place through the formation of hydrophobic interactions [103], hydrogen bonds [104,105], ionic/electrostatic interactions [106,107], crystallization, and stereo complex formation. Chemically cross-linked CS is prepared by stable binding in which polymeric chains are held together by covalent bonding through free radical polymerization, addition and condensation polymerization, enzyme induced cross-links, Diels–Alder click chemistry, Schiff base reaction, oxime formation, and Michael addition [108].

Physical cross-linking could increase the stability of the CS through interaction between cationic CS and negatively charged ionic cross-linker. It does not need the presence of catalysts or the intense purification of the final product. The process has received prominent advantages in biomedical safety due to the absence of chemical cross-linking agents, which lessen the cytotoxic effects. However, these types of hydrogels are reversible, mechanically unstable, and lack permanent junctions between the polymers, which may lead to the natural dissolution of hydrogels from the aqueous medium [109,110,111].

Compared with physical cross-linking, chemical cross-linking is a versatile method to alter the biological application characteristics of CS. Better stability and excellent mechanical properties remain as the binding between CS with the cross-linker through a covalent bond [112]. It also displays irreversible and permanent junction behavior within the hydrogel system. In addition, chemically cross-linked CS-based scaffolds present resistance to environmental variables. However, several concerns accompany this preparative technique, including adverse effects of toxic chemical cross-linking agents and difficulty in sterilization [113]. The enzymatic cross-linked process can often be controlled by adapting the temperature, pH, or ionic strength. The enzymes are mostly active under mild aqueous reaction conditions. However, the cross-linker is expensive, and the process remains substrate specific.

Cross-linked CS is developed by one-step cross-linking of functional groups with cross-linkers, while structure-modified CS is prepared through chemical functionalization [114,115]. The active chemical properties of C6–OH and C2–NH2 can be used as handles for functionalization to create CS derivatives through various kinds of molecular design. These CS derivatives can improve the physical and chemical properties of pure CS. Structure modification is the primary method of improving the water solubility of CS by introducing a hydrophilic group to an amino group. The original hydrogen bond and crystallinity of CS will be destroyed. Then, various kinds of CS derivatives appear. CS quaternary ammonium salt has been widely used as nontoxic or low-toxicity antibacterial material [116]. The pH- and thermo- sensitivity can be increased by introducing a sensitive acyl group in CS to control a drug’s release in a delivery system [112,117,118]. CS derivatives with excellent properties have been synthesized through chemical reactions and have also broadened the scope and fields of its applications in BTE [119,120]. The advantages and limits of various modified methods of CS are compared in Table 3.

Graft copolymerization between CS and other polymers is another modification strategy in BTE [121,122]. The main copolymerization methods are physical interactions [123] and graft polymerization. Different methods have their own main characteristics [124]. Graft copolymerization is a simple method [125] to improve native properties of CS [126] such as enhancing complexation or chelating properties and antimicrobial and bacteriostatic effects [127]. This process alters the surface properties, while the modified product still retains the bulk properties of CS [128]. The use of grafting reduces desorption and conveys long-term chemical stability because of its covalent nature. The graft copolymerization of CS copolymer holds great promise for widespread use in producing sustained-release drugs and other biopharmaceuticals for BTE.

## 5. Applications of Chitosan Cross-Linking Modification for BTE

### 5.1. Physically Cross-Linked Chitosan for BTE

Physically cross-linked hydrogel is one of the ways to prepare CS-based scaffolds, which are typically created by secondary forces. The polymeric network chains are formed via non-covalent bond interactions. Physical cross-linking is a ‘green’ and environmentally friendly method to enhance the functions of CS-based scaffolds [129] and has prominent advantages in biomedical applications [130,131,132]. A new CS physical hydrogel with various degrees of deacetylation (DDs) was prepared through diverse DDs of nanoporous chitin hydrogels under mild conditions [133]. The hydrogels were transparent and mechanically robust due to the extra intra- and intermolecular hydrogen bonding interactions between the amino and hydroxyl groups on the nearby CS nanofibrils. Pan et al. set up cross-linking networks based on poly(vinyl alcohol) (PVA) and CS through direct ink writing (DWI) [134]. The cyclic freezing–thawing followed by sodium citrate solution soaking yielded the first network of PVA crystallization and the second one of CS ionic interactions between amino and carboxyl groups. PVA as a cross-linker increases the resistance of CS through hydrogen bonding with the amino group of chitosan molecules [97]. The optimized composite hydrogel has a high toughness. The evidence supporting this CS physical hydrogel induction the differentiation of mouse bone marrow mesenchymal stem cells (mBMSCs) into epidermal cells in cooperation with EGF and IGF-1 in vitro has potential for use in BTE.

In another study, gelatin, CS, and a nano calcium phosphate blend was utilized for BTE scaffolds [135]. Carboxylate groups of gelatin exhibit a negative charge when the pH of the medium is higher than 4.7. Therefore, the positively charged ammonium ions of CS could interact with carboxylate groups of gelatin, resulting in the formation of electrostatic cross-linking. A further interesting example is the formation of a hybrid CS–gelatin hydrogel, the mechanical properties of which are strongly enhanced upon the addition of phytate, a multivalent negatively charged ion, to the hydrogel [136]. This system proves that the physico–chemical properties of CS are linked to the hydrogel features. In fact, when CS is neutralized with sodium phytate, a rather dense precipitate is formed due to the high charge density and stiffness of the polysaccharide. In contrast, a well hydrated, elastic hydrogel is formed when co-crosslinked with gelatin. Finally, the composite hydrogel system presents a self-healing capacity, which originates from the dynamic nature of the ionic cross-linking point [137].

Owing to the excellent load-bearing and low-friction properties of “double network (DN) hydrogels”, extensive studies on the clinical applications of DN gels as a composite scaffold for BTE have been performed [138,139]. The DN hydrogel comprises two asymmetric network structures with differing characteristics and is stiff and robust [140,141]. The rigid network as a sacrificial bond could effectively spread energy, while the flexible network could maintain structural integrity during deformation [142,143]. This principle is considered universal, having been observed in various rigid materials, such as polymers, metals, and ceramics [144,145]. It also accounts for the toughness of natural tissues [146]. Bi et al. prepared a physically cross-linked PVA/CS DN hydrogel with surface mineralization (Figure 1) [147]. In the low-temperature environment, the PVA molecular chain could form the crystalline region and interact with CS chains to form stable hydrogen bonds, heightening their thermal stability [148]. However, a significant challenge is to fabricate conductive hydrogels with high stretchability, excellent toughness, outstanding sensitivity, and low-temperature stability [149]. Therefore, a type of conductive hydrogel consisting of a DN structure is synthesized. The dynamically cross-linked CS and the flexible polyacrylamide network doped with polyaniline constitute the DN through the hydrogen bonds between the hydroxyl, amide, and aniline groups [150]. The flexible electronic sensors based on the DN hydrogels demonstrate superior strain sensitivity and linear response to various deformations. The large amount of fracture energy absorbed by CS contributed to the perfect mechanical properties. The hydrogel is cytocompatible, nonhemolytic, and suitable for bone repair.

### 5.2. Chemically Cross-Linked Chitosan for BTE

To add new functionalities and adapt the mechanical properties of the scaffolds for BTE to the desired needs, the formation of hybrid chemically cross-linked CS-based hydrogels has been extensively probed. The strong adhesive, anti-inflammatory, hemostatic, and bactericidal properties of CS make this polysaccharide an excellent candidate for a broad range of BTEs. We point the reader to some reviews on the topic.

#### 5.2.1. Aldehyde

Aldehyde is a covalent cross-linking agent for CS. The free aldehyde groups perform a Schiff reaction with the amino groups of CS. In the field of natural and/or synthetic polymer preparation and stabilization, glutaraldehyde (GA) is the most commonly used cross-linker for CS [151]. Studies proved that the addition of GA to the scaffold composition reinforces the mechanical properties and further incorporates ceramic granules, which, besides their bioactivity, will facilitate cell adhesion by the creation of contact points over the scaffold surface. Y.Z. Zhang et al. reported that cross-linking improved the mechanical performance of the gelatin fibrous membrane when the electrospun gelatin nanofibers were cross-linked with glutaraldehyde [152]. Compared with those of the untreated membrane, the tensile strength and modulus of the cross-linked membrane increased nearly ten times. Rosana V Pinto et al. [153] developed composite scaffolds, based on cross-linked CS with GA, combined with different atomized calcium phosphates (CaP) granules-HAp or biphasic mixtures of HAp and β-tricalcium phosphate (β-TCP). The biological assessment of the composite scaffolds showed that the specimens with 0.2% crosslinking were the ones with the best mechanical behavior and osteoblastic biocompatibility, as well as the highest osteogenesis-related gene expression, as shown in Figure 2. However, the toxicity [154] and legislative issues [155] of GA have restricted its application thus far. In comparison, the potential excess of free dialdehydes in the scaffold structure will compromise the cellular behavior [153] due to the cytotoxicity and side effects of glutaraldehyde exposure, e.g., asthmatic symptoms, rhinitis, and skin irritation. Researchers are committed to finding and developing new aldehyde cross-linking agents.

Vanillin (4-hydroxy-3-methoxy benzaldehyde), the main component of vanilla bean extract, has emerged as a promising natural, nontoxic cross-linking agent for CS [156,157]. The shear-thinning characteristic and improved mechanical properties of CS cross-linked by vanillin have long been known and have potential in the 3D printing technique for BTE [158]. Zhang et al. cross-linked CS films with vanillin, and the results proved the beneficial effect of vanillin on the mechanical properties of CS films [159]. The CS films cross-linked with 5% vanillin produced a 1.53-fold increase in their tensile strength from 6.64 MPa, in the case of the uncross-linked CS films, to 10.18 MPa. Limei Li et al. fabricated a featured resveratrol (Res) delivery nano-hydroxyapatite (n-HAp)/CS composite microsphere cross-linked by vanillin/ethanol solution [160]. The microspheres had anti-inflammatory activity evidenced by the decreased expression of pro-inflammatory cytokines TNF-α, IL-1β, and iNOS in RAW264.7 cells in a dose-dependent manner. The composite microspheres could also stimulate BMSC proliferation and osteo-differentiation, as well as enhance entochondrostosis and bone remodeling under osteoporotic conditions. In addition, cinnamaldehyde, an eco-friendly bactericidal agent, might be used as a promising cross-linker for preparing CS particles due to its aromatic conjugation and aldehyde group. Further, as an antimicrobial agent, it could improve the stability and antibacterial properties of CS particles, making them a green solution with a wide range of applications in life sciences and BTE [161].

#### 5.2.2. Genipin

Genipin is a natural cross-linking agent extracted from the gardenia plant [162]. It is significantly less cytotoxic than GA and has superior antibacterial properties, reducing the likelihood of bacterial adherence following scaffold implantation [163,164]. The mechanism of genipin cross-linking is pH-dependency. Under acidic and neutral circumstances, the amino groups of CS react with the olefinic carbon atom at C-3 of genipin to open the dihydropyran ring [165] and create heterocyclic amines. Genipin occurs through the nucleophilic attack by hydroxyl ions in an aqueous solution to form intermediate aldehyde groups, which subsequently undergo aldol condensation. Similar to aldehydes, the aldehyde groups of genipin can react with the amino groups of CS to form the Schiff base and then create a net-like structure [166]. BTE scaffolds made up of genipin-cross-linked CS have been extensively studied in terms of manufacturing and characteristics. Researches have proved that osteoblast-like MG-63 cells could be cultured on genipin-cross-linked CS scaffolds [167]. Wu F et al. loaded a genipin cross-linked carboxymethyl chitosan (CMCS) hydrogel with gentamycin and achieved increased adhesion, proliferation, and differentiation of osteoblasts as well as full inhibition of *Staphylococcus aureus* [168] (Figure 3). Genipin also enormously improves the mechanical properties of composite scaffolds. Genipin can form intramolecular and intermolecular cross-linking networks and can further form an interpenetrable polymer network (IPN) [169]. During the cross-linking step, more cross-linking points will lead to higher biostability. For instance, the elastic modulus of the genipin-cross-linked CS-PVA blend increased from 0.22 to 2.08 MPa. Nevertheless, this value was lower than when GA was employed, but genipin is much less cytotoxic than GA [170]. Thus, it would be safer to employ the naturally procured genipin to augment the mechanical properties of CS [171]. E. Frohbergh et al. developed a one-step platform for electrospun nanofibrous scaffolds of CS, which also contained HAp nanoparticles and were cross-linked with genipin [172]. The results showed that the cross-linked CS/HAp scaffold resulted in a five-fold increase in Young’s modulus, approximating that of periosteum. Rheological studies by Pandit et al. showed that the stiffness of hydrogels made of methylcellulose, CS, and agarose increased upon cross-linking the CS with increasing amounts of genipin [173]. In fact, the growth of osteoblasts and the differentiation marker expression of osteoblasts will be both enhanced in cross-linked CS gel and demonstrate significant bacterial inhibition.

There is also evidence proving that genipin can be exploited in bioactive scaffolding systems able to finely tune the role of inflammatory cells towards a regenerative phenotype while avoiding chronic inflammation and the resulting fibrotic capsule [174]. Simona Dimida et al. used human monocyte-like THP-1 cells to evaluate their inflammatory morphological responses towards the suspension–adhesion transition under treatments with Phorbol-12-myristate-13-acetate (PMA) on a genipin-cross-linked CS scaffold [175]. Evidence from monocyte-like cells showed that the genipin seems to promote the slowing of the monocyte-macrophage transition at a morphological level.

CS scaffolds cross-linked with genipin are most attractive for BTE. While genipin is expensive because a large quantity is wasted during its preparation due to homopolymerization, minor amounts of genipin are necessary. Today, genipin is only used in experimental studies, and there is no economic justification for its use in mass production.

#### 5.2.3. Tripolyphosphate (TPP)

TPP (Na_5_P_3_O_10_), an ionic cross-linker [176], is the salt sodium penta-anion polyphosphate and the conjugate base of triphosphoric acid. TPP interacts with the amino groups of long polymer molecules and forms a 3D network of ionically cross-linked regions. Then, the system will be inhomogeneous. The reaction is strongly associated with pH [177]. This process is more straightforward and gentler than previous cross-linking methods [178,179]. Moreover, since phosphate groups are considered necessary for bone mineralization, TPP is also widely employed as a cross-linker to develop biomimetic polymer systems for bone regeneration [180] and can enhance the mechanical properties suitable for human bone. For pure CS film, TPP cross-linked film significantly improved the elastic modulus compared to the noncross-linked one, whose elastic modulus values were close to the reported values for human bone [181], but more brittle [182]. For composite CS-based biomaterials, cross-linking CaP/CS paste using the TPP solution significantly increased the strength and Young’s modulus, much more noticeably in the wet state of the CaP/CS scaffolds [183]. The TPP concentration also affects the scaffolds’ dimensional stability in aqueous medium and the mechanical properties [184,185,186]. There are studies reporting that scaffolds treated with TPP at concentrations from 2.5 to 5% had good physical and structural integrity. A higher TPP cross-linking concentration and more time enable the creation of stable cytocompatible scaffolds for 3D anisotropic tissue formations. Osteoblast adhesion and vitality can also be enhanced by simulating the structure of mineralized cortical bone by incorporating TPP-cross-linked CS and bioceramics, such as nHAp and β-TCP. Simultaneously, the cross-linking reaction can favor porous structure formation with convenient features for application in BTE [187]. The porous structures are conducive to nutrient transport and host blood vessel construction. Suren P Uswatta et al. [188] fabricated porous injectable spherical nHA/CS scaffolds via non-toxic coacervation and lyophilization techniques (Figure 4). TPP-cross-linked CS could promote osteoblast adhesion, and the addition of nHA increased the ultimate tensile strength of the scaffold.

However, the simple mixing of CS and TPP solutions is not suitable for regulating the cross-linking reaction since instantaneous gelation leads to precipitate formation. With better and standardized fabrication techniques, the potential clinical use of a cross-linking agent could lead to improved bone substitute and tissue engineering applications. However, other important factors such as biodegradability and biocompatibility should also be considered carefully. Therefore, different approaches should be employed to prepare suitable scaffolds for BTE [189,190].

#### 5.2.4. Other Cross-linkers

As previously stated, hydrogel materials are in constant use for their excellent biological properties, while improving their processability and mechanical properties is still required. To address the present constraints, Ana Mora-Boza et al. reported the fabrication of dual cross-linked 3D scaffolds using a low concentrated (<10 wt%) ink of gelatin methacryloyl (GelMA)/CS with a novel cross-linking agent, glycerylphytate (G1Phy) [191]. G1Phy is a hybrid derivative of phytic acid with reduced toxicity. The ionic post-treatment mediated by G1Phy provides fast and homogeneous ionic cross-linking between phosphate groups in G1Phy and the amine groups in CS and GelMA. The cross-linking process is crucial for the long-term stability properties of the polymeric networks. The preliminary in vitro testing with L929 fibroblasts revealed the encouraging adhesion, spreading, and proliferation outcomes when compared with other phosphate-based conventional crosslinkers (e.g., TPP) used for BTE.

Zwitterionic CS is an environmentally benign biomaterial of interest [192]. Ionic cross-linking of polycationic CS with carboxyl anions renders a polyampholytic (mixed charged) character to the polymer. Paulomi Ghosh et al. evaluated the fiber-forming ability of CS with citric acid at physiological pH via instantaneous ionotropic complexation [193]. The citrate–CS fibers were further cross-linked via carbodiimide chemistry to introduce amide bonds in the network structure, forming a dual cross-linked network. The dual cross-linked fibers demonstrated superior protein adsorption and bio-mineralization among the fiber types, giving rise to higher mesenchymal stem cell adhesion and better osteogenesis. Furthermore, in vivo experiments confirmed the mechanical stability and osteoconductive nature of the dual cross-linked fibers.

#### 5.2.5. Photo-Cross-Linked Chitosan

As an advancing technology, the digital light processing (DLP) technique employs a digital mask projection to trigger localized photopolymerization. The method enables high-efficiency fabrication of 3D hydrogel structures with high precision ranging from 1 to 100 μm, which plays an essential role in fabricating unique 3D objects in the biomedical fields [194,195]. The easy availability of DLP printers has made this technology promising, primarily in personalized medicine [196]. The key to photocurable printing is photocuring hydrogels [197]. Until now, photo-crosslinked CS has been extensively used for the DLP of BTE.

The most frequently used method of photo-cross-linking CS is the free-radical polymerization of (meth)acrylate-based monomers. Photo-radiation produces free radicals by dissociating photoinitiators that are added to the bioink. The chitosan methacryloyl (CS–MA) polymer networks are formed through photopolymerization of carbon –carbon double bonds between CS molecular chains. The higher the DS, the more dense the intermolecular cross-linking degree. More importantly, through DLP-based 3D printing, the optimized CS–MA can be processed into complex 3D hydrogel structures with rapid and accurate spatiotemporal control, high-resolution, high-fidelity, and good biocompatibility [198]. The aggregation of CS–MA and graphene oxide (GO) is favorable for bone regeneration. The platelets promote the migration and proliferation of osteogenic cells, increase blood vessel formation, and induce inflammatory reactions [199].

Recently, bioorthogonal click reactions such as thiol–ene click chemistry have raised considerable attention as an alternative cross-linking mechanism to chain-growth polymerization. These reactions can proceed via Michael-addition reactions or a step-growth polymerization under light irradiation (mostly UV or visible light). This cross-linking method has three steps: initiation, propagation, and termination. Zhou et al. synthesized photo-clickable thiol-ene hydrogels based on CS using photopolymerization of maleic chitosan (MCS) and thiol-terminated PVA in the presence of a biocompatible photoinitiator (Figure 5) [200]. MCS was synthesized by the ring-opening reaction of CS with maleic anhydride, and photopolymerized MCS/PVA hydrogels were obtained by thiol-vinyl photopolymerization. Studies found that as the PVA content increased, the MCS/PVA hydrogels presented a higher compressive modulus, slower absorption rate, lower equilibrium swelling ratios, and more compact pores, suggesting an increased cross-linking density of the hydrogel network. The properties promoted L929 cell adhesion and proliferation on the hydrogel surface.

### 5.3. Enzymatic Cross-Linked Chitosan for BTE

In general, enzymatic cross-linking methods have been applied to synthesize various polymeric platforms and hydrogels. The primary enzymatic cross-linkers are transglutaminases (protein glutamine gamma-glutamyltransferase), tyrosinase (Tyr), lysyl oxidase, phosphatases, and horseradish peroxidase (HRP), and hydrogen peroxide (H_2_O_2_). Enzymatic cross-linking provides high reaction rates under physiological conditions. It is also a “green” approach with mild reactions and biocompatible catalysts for hydrogel synthesis [201]. Substitution of phenol-containing functional groups on CS is a widely explored method for developing injectable hydrogels through HRP-mediated enzymatic cross-linking [202]. HRP-mediated cross-linking of proteins and peptides through the tyrosine residues and growth factors, such as bone morphogenetic proteins-2 (BMP-2), contains multiple tyrosine residues within the peptide sequence. The oxidative tyrosine coupling reaction of BMP-2 can lead to covalent binding of the protein to 3-(4-Hydroxyphenyl)propionic acid (HPP)-modified CS chains during cross-linking. Shalini V Gohil used a critical-sized bilateral calvarial defect model to compare the osteogenesis potential of human bone morphogenetic protein-2 (rhBMP-2)-loaded enzymatically cross-linkable HPP-modified glycol chitosan (HRP-GCS + BMP) and the collagen HAp matrix “Healos^®^”. The spatial control of rhBMP-2 bioactivity at the cellular level was confirmed by fluorescence expressed in osteoblast and pre-osteoblast cells. The retained rhBMP-2 in the HPGC + BMP implant could localize osteoprogenitor recruitment and osteogenesis while also minimizing rhBMP-2 diffusion loss at the implantation site (Figure 6).

In addition, the enzymatic crosslinking reaction produces large quantities of entangled nanofibers, contributing to a denser network and smaller pore size of the composite gel network [203], which can offer better mechanical properties and excellent chemical and thermal stability compared with ionically crosslinked polymer networks [204]. Studies are in progress to develop an enzymatically cross-linkable hydrogel platform with good spatial localization of the encapsulated growth factor and controlled, degradation-dependent release characteristics.

Overall, cross-linked CS’s rigidness, acid-solubleness, reusability, and selectivity have shown more remarkable improvements than pure CS in BTE. Table 4 provides the cross-linking mechanism of different cross-linking agents and CS for a clearer understanding.

## 6. Application of Structure-Modified Chitosan for BTE

Another significant concern in developing CS biological properties for BTE is the CS derivates. Recently, there has been a growing interest in structure modification of chitosan to improve the solubility of these compounds and widen their applications in tissue engineering [205]. Given that the primary and secondary hydroxyl groups at the C-6 and C-3 locations of CS are active functional groups that are susceptible to chemical reactions, the structure could be modified by being acylated, esterified, alkylated, etherified, azidated, and halogenated. CS and its derivatives are natural polymers that exhibit enzymatic biodegradability, pH sensitivity, a polycationic nature, etc. [206]. Several studies showed that CS with substituent incorporation can decrease intracellular reactive oxygen species (ROS), thereby boosting the osteogenic differentiation of mesenchymal stem cells (MSCs) [207]. Scaffolds based on CS derivatives have favorable biocompatibility and physicochemical properties, which hold much promise for BTEs.

### 6.1. Carboxymethyl Chitosan, CMCS

Inspired by the natural extracellular matrix, carboxymethyl chitosan (CMCS)-based composite scaffolds have great potential in BTE. CMCS is an anionic CS derivative. Modification of CS by carboxymethylation of hydroxyl and amino groups affords better solubility of aqueous solution at different pH values, which achieves the processability in BTE. Chelation with more Ca2+ is enabled due to the introduction of carboxymethyl groups in CMCS [208]. Furthermore, the enhanced mineral deposition could be attributed to the carboxymethyl groups for providing more nucleation sites [209,210], also effectively regulating the nucleation and growth of apatite from the solution [211]. In addition, CMCS enhanced the biodegradability after introduction of carboxymethyl [212,213]. Studies have shown that CMCS electrospun scaffolds promoted proliferation, and suitable cell–cell and cell–environment interactions were observed, especially at maximum concentrations of CMCS [214]. The research of Zhang et al. also proved this view [215]. In physiological circumstances, several functional chemical groups of CMCS stimulated osteoblast adhesion, proliferation, and CaP deposition (Figure 7). There are also studies focusing on the analgesic and anti-inflammatory properties of O-CMCS by applying classical rat pain and inflammation models [216]. Results of anti-inflammatory properties show that O-CMCS inhibited inflammation induced by carrageenan in the hind paw of rats, which demonstrated that O-CMCS has remarkable anti-inflammatory activity.

The improvement in compressive strength makes it an attractive candidate for bone defect repair at the same time. Because of its porosity, gel-forming characteristics, simplicity of chemical manipulation, and high affinity for in vivo macromolecules, CMCS will be a viable contender as a supporting material for BTE [217].

### 6.2. Hydroxypropyltrimethyl Ammonium Chloride Chitosan, HACC

Bone substitutes exhibiting osteoconductivity and antimicrobial activity are increasingly needed to prevent and treat contaminated or infected bone defects. Hydroxypropyltrimethyl ammonium chloride chitosan (HACC) is a water-soluble CS derivative with a strong cationic nature. The addition of quaternary ammonium groups to the CS molecule preserves the characteristics of raw CS and significantly weakens hydrogen bonds, improving water solubility and antibacterial activity [218]. Foremost, HACC is biodegradable, is used to treat multidrug-resistant bacterial infections, and is successfully used as an antimicrobial agent for BTE. The antimicrobial activity and biocompatibility of HACC could be adjusted by varying the degree of substitution (DS) of quaternary ammonium. Studies have proved that HACC with 26% DS displayed a significantly enhanced antibacterial effect over that of the CS and glycidyl trimethylammonium chloride (GTMAC) [219]. The electrostatic interaction between the positively charged quaternary ammonium groups of HACC and the negatively charged phosphoryl groups of the phospholipid components of bacterial membranes affected the cytoplasmic membrane integrity [220], thereby inhibiting bacteria from forming biofilms and enhancing the bone regeneration properties effectively [221].

Antibiotic-loaded polymethyl methacrylate (PMMA) has successfully treated and prevented osseous infections. It represents the current gold standard for local antibiotic delivery systems in orthopedic surgery [222]. However, the local overuse of antibiotics also leads to the evolution of antibiotic-resistant bacteria, which accounts for the failure of anti-infective treatments [223]. To solve the situation, Tan et al. reported that HACC, which was loaded into PMMA, significantly inhibited the formation of biofilms caused by methicillin-resistant *Staphylococcus* strains [224]. Further studies showed that HACC-loaded PMMA could improve properties, e.g., a lower polymerization temperature, prolonged setting time, higher hydrophilicity, greater apatite formation on the surface after immersion in simulated body fluid, and better attachment of the hBMSC, which is a better choice for better osteointegration in BTE (Figure 8). The good biocompatibility, antibacterial nature, and osteogenic activity displayed by the HACC-grafted scaffold makes it a potential option for the regeneration of contaminated or infected bone defects.

### 6.3. Sulfated Chitosan, SCS

Another CS derivative is SCS. It is the general term for sulfonated and sulfated CS derivatives. SCS is the production of sulfonation reaction, the incorporation of sulfonate groups onto CS [225]. Due to the existence of residual amino groups, the resulting SCS chains present polyampholytic characteristics encountered in the structure of some sulfated GAGs, a particular class of complex charged polysaccharides involved in the extracellular matrix (ECM) (e.g., chondroitin sulfate, heparin). GAGs are known to regulate cell behaviors, such as cell adhesion, migration, proliferation, and differentiation [226,227]. Thus, SCS derivatives should benefit from these excellent biological properties depending on the degree of substitution.

The most attractive characteristic of SCS is that it has been employed for its blood anticoagulant properties ascribable to their chemical structure similar to that of heparin [228,229]. SCS could act as heparan sulfate mimetics to regulate protein growth factor activity and other physiological processes. The stimulation from 6-O-sulfated chitosan (6SCS) could enhance the bioactivity of BMP-2. The increased chai length and further sulfation on 26SCS also resulted in higher ALP activity. Research indicated that BMSCs cultured in the coculture system of N, 6-O-Sulfated chitosan (26SCS), and BMP-2 exhibited higher cell viability. Further, more vascular endothelial growth factor (VEGF) and NO were secreted to improve the angiogenic potential of BMP-2 and thus could lead to better bone regeneration [230] (Figure 9). Han et al. elucidated the effects of SCS coating on the poly(d,l-lactide) (PDLLA) membrane on the HUVEC and MC3T3-E1 co-culture system [231]. It was suggested that SCS could influence the bone repair microenvironment and stimulate osteoblast proliferation and activity by upregulating gene expression and supporting micro-angiogenetic processes via new bone formation. Furthermore, various concentrations of SCS have different effects on the osteogenic activity of BMP-2. A low dose of 26SCS enhances BMP-2-induced mineralization in vitro and ectopic bone formation in vivo. Additionally, a low dose of 26SCS promotes the interaction between BMP-2 ligands and receptors and inhibits the function of Noggin [232]. Hence, SCS is a more potent enhancer of BMP-2 bioactivity than native heparin, which is promising for future applications in BTE as a synergistic factor of BMP-2 for local bone regeneration.

### 6.4. Glycol Chitosan, GCS

Glycol chitosan (GCS) is a water-soluble derivative of CS wherein the C6 hydroxyl groups are functionalized with glycol groups [233]. It is ideal for producing biomaterial, which is soluble under alkaline, neutral, and acidic conditions, that is, over the whole pH range studied [234]. GCS and its nanoparticulate formulations are regarded as excellent drug delivery vehicles due to their appropriate biocompatibility, biodegradability, and bioadhesive nature [235]. GCS can also be further developed into hydrogel scaffolds for BTE. GCS ligands are likely to stabilize the formed nHA inorganic cores by creating an organic shell with the hydrophilic glycol moieties preferentially oriented toward the water medium and providing steric hindrance that prevents nanoparticle aggregation and agglomeration, which is different from the electrostatic stabilization by carboxylic groups reported for carboxymethyl CS [209,210,236]. The morphology of the cells shows predominantly elongated sprawling on membranes emitting cytoplasmic processes that facilitate adhesion and cell communication [237,238], which also suggests the biocompatibility of the HBMS cells.

GCS is also an excellent drug carrier for delivery systems owing to its physicochemical features, biocompatibility, biodegradability, and mucoadhesiveness [239]. In the study of Chih-Wei Chiang et al. [240], SrR, used for the treatment of osteoporosis, was added in GCS/HA nanoformulation through electrostatic interaction. A cell viability test demonstrated SrR nanoparticles (SrRNPs) exerted no cytotoxic effects on osteoblasts in vitro. Radiographical and histological analyses suggested a higher level of bone regeneration in the SrRNPs-H-implanted groups than in other experimental groups. Interestingly, the hydrogel carrier promoted local site-effective delivery of SrR. GCS is expected to provide enough mechanical support for a steady release of drugs to guarantee drug activity and safety in BTE.

### 6.5. Guanidinylated Chitosan, GC

Guanidine is an essential class of organic compounds. It is present in many medicinally important compounds that show biological activity and therapeutic functions, such as broad-spectrum antimicrobials and antidiabetic drugs. Polymers functionalized with amine groups can be easily converted to guanidinium groups. Attachment of the guanidine group onto CS can introduce a positive charge onto the polymer backbone, which would result in better aqueous solubility at neutral pH and shows potentially good antimicrobial activity [241]. Surface hydrophilicity and hydrogen interactions would increase to improve self-healing, shape memory, and mechanical and biological properties by loading this semi-conductive CS derivative into the polymer structure. Research showed that GC could stimulate the osteoblast differentiation of mesenchymal stem cells and upgrade the mineralization process [242]. The presence of the positively amino and guanidino groups will create a specific signal for cell recognition and attachment (Figure 10). There are also studies focusing on the effect of GC on osteogenic signaling pathways. Zhang et al. developed a self-healing and pro-osteogenic hydrogel system based on the self-assembly of laponite nanosheets and guanidinylated CS [242]. This hydrogel system offers a multi-functional encapsulation platform for encapsulating living cells, therapeutic agents, and synthetic bone grafts structure. Simultaneously, it may stimulate the Wnt/-catenin signaling pathway, which promotes cell adhesion and osteogenic differentiation in mesenchymal stem cells to enhance bone regeneration.

There are also some other structure-modified strategies of CS. It has been modified with 5-fluorouracil to serve as an in vitro anti-tumor drug delivery system [243]. It has also been modified with different simple organic and polymeric materials, especially with halloysite nanotubes (HNTs). HNTs are unique inorganic nano clays formed by aluminosilicate kaolin sheets that are rolled. HNTs have been used successfully in several studies as nanomodifiers [244,245,246]. This is an attractive option for BTE due to boosting the mechanical properties of polymer matrix resulting from their high aspect ratio and tough structure and the absorbable degradation products that induce osteogenic cell differentiation [247]. Several studies have proved the favorable biological properties of the scaffolds and hydrogels consisting of CS and HNTs [98,248,249]. Liu et al. combined solution-mixing and freeze-drying techniques to develop novel CS/HNT nanocomposite (NC) scaffolds [250]. Measurements of the NC scaffolds’ mechanical and thermal properties revealed a substantial improvement in compressive strength, compressive modulus, and thermal stability compared with those of the pure chitosan scaffold. Meanwhile, adding a significant amount of HNTs (up to 50 wt.%) to the biopolymer matrix did not disturb cell proliferation and hemocompatibility [246]. Kadam AA et al. designed a surface-engineered nano-support for enzyme laccase-immobilization through grafting the surface of HNTs with Fe_3_O_4_ nanoparticles and CS [251]. These CS-based, rapidly separable super-magnetic nanotubes for efficacious enhancement of laccase biocatalysis can be applied as nano-supports for other enzymes. In conclusion, HNT-modified CS can greatly improve physical defects while maintaining the original biological properties of the CS, which shows excellent potential in BTE and can be used as stem-cell carriers for bone repair [252].

## 7. Application of Chitosan Grafted with Biodegradable Polymers for BTE

During the last decades, CS has been widely modified by graft copolymerization with a multitude of polymers to improve some CS weaknesses to broaden its application in BTE [253], such as the limited chain flexibility, the poor mechanical strength, the low thermal resistance or its low selectivity as an adsorbent [254]. Generally, a graft copolymer consists of a high-weight macromolecular chain of one monomer, referred to as the backbone polymer, with one or more branches or grafts of different monomers/polymers [255,256]. CS usually acts as the backbone chain in grafting due to its high molecular weight [257,258]. The amino and hydroxyl groups are the points of initiation of the graft copolymerization [259]. The biodegradable polymers that have been copolymerized with CS are very varied, and the most studied include PLA, polycaprolactone (PCL) [260], lignocellulosic products, pectin [132], gelatin [261,262,263], silk proteins [264,265], and peptides [266,267]. Cui et al. grafted CS on electrospun PLA nanofibers to induce the deposition and growth of HA crystals [268]. The HA content and growth kinetics can be modulated by changing the CS content grafted on the surface of electrospun fibers. The mineralized scaffolds from CS-grafted PLA fibers provided favorable conditions for the proliferation of MC3T3-E1 cells, and they have the potential to be used in applications as coating materials on medical devices and as scaffolds for BTE. The in vitro fibroblast static cultivation on CS/PLA films has also a faster cell growth rate compared to CS [269]. Hydroxypropyl methylcellulose (HPMC) could be grafted with CS and, subsequently, complexed with carboxymethylcellulose (CMC) to obtain a polyampholytic hydrogel for sustained drug delivery [270]. In addition, CMC can establish strong ionic cross linkages with CS to form polyelectrolyte complexes. The composite scaffolds are cyto-friendly and can promote differentiation of mMSCs to osteoblasts, which is useful as scaffolds to develop bone tissues [271].

Furthermore, developing hybrid biomaterials by including peptide sequences into biopolymers is an attractive alternative to confer functional cell–biomaterial interaction [272]. RGD peptide and HVP-aldehyde peptide can also be grafted with CS. In the study of Wang et al. [273], a cyclic RGD peptide was grafted with CS by a thiolation reaction and a cross-linking agent and was used in addition to GO in drug delivery applications. Paola Brun et al. reported that the 1:1 mixture of HVP functionalized-CS:CS is the best compromise between preserving the antibacterial properties of the material and supporting osteoblast differentiation and calcium deposition [ES]. The published literature also shows that grafted chitosan is a promising substance for biomedical applications. In the future, the production and clinical use of new graft copolymers of chitosan will provide a new approach to BTE research of commercial importance.

Overall, pure CS could be structure-modified through amino and hydroxyl groups such as acylation, carboxylation, etherification, graft copolymerization, and ring opening of epoxides to produce a series of CS derivatives (Table 5). These CS derivatives show improved water solubility, biological activity, and mechanical properties compared to native CS, which would be more appropriate in the applications of BTE.

**Table 1 ijms-23-06574-t001:** Summary of the following different types of modifications of chitosan, fabrication techniques, bioactive molecules, and experimental model system in the study of bone regeneration in vitro and in vivo.

Modification	Fabrication	Materials	Effect	Cell/Model	Ref.
Physically cross-linked	ice template-assisted freeze-drying	EO-loaded CS/Dex	Exhibit antioxidant, antifungal properties and the inhibition of *Candida parapsilosis* fungi	-	[129]
	freeze-drying	nanoporous chitin hydrogels	Enhance the strength and Young’s modulus of hydrogel, mBMSC adhesion, and proliferation	mBMSC	[133]
	direct ink writing (DWI)	CS/PVA	Promote toughness performance	-	[134]
	double network	CS/PVA/HAp	Increase cell adhesion, proliferation, OCN, ALP, COL I, and osteochondral repair efficacy	Rat bone marrow mesenchymal stem cells (rBMSCs) and L929 cells (Mouse fibroblast cell line)/New Zealand white rabbits with a bone defect (5 mm in diameter and 8 mm deep) in the lateral femoral condyle	[147]
Aldehyde-crosslinking	freeze-drying	CS/HA/β-TCP	Promote biological performance, metabolic activity, ALP expression, cell morphology, cell/scaffold interaction, and gene expression	MG63 human osteoblastic-like cells	[153]
	freeze-drying	CS/vanillin hydrogel	Achieve a good balance between self-healing capability and mechanical strength	-	[156]
	emulsion method	CS/vanillin hydrogel	Provide favorable cell attachment and biocompatibility	MG63 cell/ muscular incision 20 mm long on the backs of SD rats	[157]
	freeze-drying	vanillin-CS/CS	Exhibit suitable viscosity values and shear thinning behavior for 3D printing applications	-	[158]
	freeze-drying	Cinnamaldehyde/CS	Show thermal characteristics and stability and synergistic antibacterial activity against *Staphylococcus aureus* and *Escherichia coli* bacteria	*Staphylococcus aureus* or *S. aureus* (ATCC 25923) and *Escherichia coli* or *E. coli* (ATCC 35218) bacteria	[161]
Genipin cross-linking		CS and hyaluronic acid solutions PEC+BMP-2	Control the swelling ratio and degradation of PEC and achieve quite a high loading efficacy, prolonged, and sustained BMP-2 release profile	MC3T3-E1 cells	[165]
	mixing	gentamycin sulfate (GS)-loaded CMCS hydrogel	Achieve superb inhibition of bacterial growth and biofilm formation of *Staphylococcus aureus*, enhance the adhesion, proliferation, and differentiation of MC3T3-E1 cells	MC3T3-E1 cells	[168]
	electrospinning	CS/HA nanofibers	Increase in Young’s modulus and osteoinductive bioactivity	Murine 7F2 osteoblast-like cells	[172]
	mixing	CS/methylcellulose	Enhance fibroblast, endothelial, and osteoblast proliferation and adhesion	Osteoblasts, fibroblasts, and HUVECs	[173]
	DWI and freeze-drying	HA/CS composite scaffolds	Friendly environment, increase cell population, levels of viability, and attachment	MG63 human osteoblast-like cells	[274]
	self-assembly	HA/GO/CS composite hydrogel	Improve the microstructure and mechanical strength. Balance the rigidity and toughness of the composite hydrogel	Rat bone marrow mesenchymal stem cells (rBMSCs)	[275]
Tripolyphosphate (TPP) cross-linking	coacervation and lyophilization	nHA/CS/TPP Scaffolds	Exhibit highest ultimate compressive strength and show good osteoblast adhesion and proliferation	OB-6 line cell	[188]
	freeze-drying	CS/Gel/β-TCP scaffolds	Show mechanical improvements, bioactivity, high proliferation rate, high extracellular calcium deposition, excellent cell adhesion, and characteristic osteoblast cell morphology	Human osteoblast cells (CRL-11372) (hOB)	[179]
	freeze-drying	HA/β-TCP/CS composites	Show good swelling properties, and higher levels of cell proliferation and growth	Human osteoblast-like cells (Saos-2) and mouse fibroblastic-like cells (L929)	[276]
Glycerylphytate (G1Phy)	3d-printing and photopolymerization	GelMA/CS scaffold	Exhibit excellent shape fidelity, resolution, swelling behavior, and mechanical and biological properties; enhance cell adhesion and proliferation	L929 fibroblasts	[191]
Carbodiimide and citric acid	extruded in a coagulant bath using viscose-type stainless steel spinneret	citrate–CS fibers	Improve the mechanical property, higher stability against enzymatic degradation and hydrophobicity, and superior bio-mineralization	MSCs New Zealand white male rabbits	[193]
Photo-crosslinking	UV light	MCS/TPVA (Darocur 2959)	Exhibit rapid gelation behavior, improved stiff and compressive strength. Promote L929 cell attachment and proliferation	L929 cell	[200]
	visible blue light with riboflavin	CS-MTT hydrogel	Recruit native cells and promote calvarial healing without the delivery of additional therapeutic agents or stem cells	male CD-1 nude mice	[277]
	blue light (420–460 nm)	ChI-MA/GO	Showe intermediate platelet aggregation hemolytic tendencies, enhance tissue regeneration	NHOst cells Reconstruction of the distal epiphysis of the femur	[199]
Enzymatic-crosslinking	Standard carbodiimide coupling method	HPP-GC + BMP(HRP + H2O2)	Localize osteoprogenitor recruitment and osteogenesis	Col3.6 rat critical sized bilateral calvarial defect model	[278]
Carboxymethyl chitosan, CMCS	electrospinning	CMCS/HA	Increase the ALP activity and Runx2 expression, promote new bone formation and maturation	mBMSCs circular critical-size Calvarial bone defects (diameter of 5 mm) on both parietal bones of Sprague–Dawley rats	[211]
	electrospinning	PCL/CMCS nanofibrous scaffolds	Adjust the viscosity and charge density and exhibited excellent initial cell attachment and proliferation	human osteoblast cells (MG63)	[214]
	freeze-drying	NOCC/FD composite hydrogel	Enhance the proliferation, ALP activity, and mineralization of osteoblast cells	L929 mouse fibroblasts and 7F2 osteoblast cell	[279]
	freeze-drying	SF/CMCS/CNCs/Sr-HAp	Maintain high porosity with a lower swelling ratio, enhanced protein adsorption and ALP activity	bone mesenchymal stem cell (BMSC)	[217]
Hydroxypropyltrimethyl ammonium chloride chitosan (HACC)	3D-printing	PLGA/HA/HACC composite scaffold	Favor cell attachment, proliferation, spreading, and osteogenic differentiation and exhibit good neovascularization and tissue integration	human bone marrow-derived mesenchymal stem cells (hBMSCs)	[219]
	solvent casting-particulate leaching method	silica/HACC/zein scaffold	Exhibit long-lasting antibacterial activity against *Escherichia coli* and *Staphylococcus aureus*, and significant early osteogenic differentiation	Rabbit model of critical-sized radius bone defect	[221]
	PTFE mould	HACC-PMMA	Improve properties, stem cell proliferation, osteogenic differentiation, and osteogenesis-associated gene expression	human mesenchymal stem cells (hMSCs)	[224]
Sulfated chitosan (SCS)	-	2-N,6-O- SCS + BMP-2	Exhibit a higher cell viability and sprouting ability, secrete more VEGF and NO, and improve the angiogenic potential	Rat bone marrow stromal cells (BMSCs)	[231]
	solution casting	SCS coated on poly(d,l-lactide) (PDLLA)	Increase osteogenic- and angiogenic-related gene and protein expression	Mouse preosteoblast cells (MC3T3-E1s) and human umbilical vein endothelial cells (HUVECs)	[231]
	-	2-N,6-O- SCS + BMP-2	Enhance BMP-2 bioactivity to induce osteoblastic differentiation in vitro and in vivo by promoting the BMP-2 signaling pathway	C2C12 cells	[232]
Glycol chitosan (GCS)	solvent cast and evaporation	nHA/GCS composites	No cytotoxicity and promotion of cell ingrowth and osteoconduction	osteoblastic-like (SAOS) and embryonic cell lines (HEK293T)	[238]
	solvent cast and evaporation	CHA/ SF/GCS/DF-PEG self-healing hydrogel + BMP-2	Promote osteogenic differentiation of mOPCs and promote the proliferation and migration of HUVECs	C57BL/6 suckling rat A 4-mm-deep hole in the femoral condyle of SD rat	[280]
	electrostatic interaction	GCS-HA NPs + PEGDA+ SrRNPs-H	Increase the level of bone regeneration	Dorsal incision around the lumbar and sacral pine area of male Wistar rats	[240]
Guanidinylated chitosan (GC)	sol–gel chemistry and freeze-drying	Sulfonate and carboxylate-containing chitosan/silica hybrid composites	Showed a substantial effect on the mineralization of calcium phosphate and was more efficient to induce heterogeneous nucleation and growth of hydroxyapatite	-	[281]
	-	GC/PANI-containing self-healing semi-conductive waterborne scaffolds	Exhibit excellent shape memory properties and shape recovery ratio, enhanced cell attachment, COL-1, ALP, RUNX2, and OCN expression	Human adipose-derived mesenchymal stem cells (hADSCs)	[282]
	mixing	LNSs/GC	Show inherent osteogenic properties, a versatile moldable vehicle, facilitating handling and osteogenic potential	Mouse bone marrow stromal cell line (BMSCs, D1 ORL UVA [D1], D1 cell, CRL-12424)	[242]
Grafted with PLA	electrospun	CS/PLA/HA	Enhance proliferation of MC3T3-E1 cells used in applications as coating materials on medical devices	MC3T3-E1 cells	[268]
Grafted with HPMC	coupling reagent-mediated approach	CS/HPMC	Highly water-soluble across a wide pH range, high pH buffering capacity, and a high drug encapsulation efficiency	Metronidazole, methylene blue, tetracycline hydrochloride, and mometasone furoate as drug models	[270]
Grafted with CMC	Freeze-drying	CS/HPMC/ mesoporous wollastonite	Cyto-friendly nature to human osteoblastic cells, confirmed by calcium deposition and expression of an osteoblast-specific microRNA	MG-63	[271]
Grafted with cycle RGD peptide	noncovalent method	CS/cRGD/GO	Provide a multifunctional drug delivery system and can be efficiently loaded with a number of therapeutic agents for biomedical applications	hepatoma cells (Bel-7402, SMMC-7721, HepG2)	[273]
Grafted with HVP-aldehyde peptide	mixing	CS/HVP	Support the adhesion of osteoblasts, the formation of elongated cell shapes, and increased osteoblast differentiation.	Human (h) osteoblast cells	[283]

## 8. Future Directions for Modified CS-Based Bone Scaffolds

The past few decades have seen revolutionary advancements for designing dynamic living constructs and replacing malfunctioned bone tissues in BTE applications. Intelligent hydrogels that respond rapidly to the exterior stimulus changes have attracted considerable attention from many scientists [284]. These “smart hydrogel scaffolds” could be modified and improve osteogenic differentiation of stem cells, leading to a more responsive reaction to changes in their surrounding environment [285], such as pH [286,287], temperature [288], light [289], and the electric field [290]. CS-based stimuli smart hydrogels have emerged as powerful platforms that provide sophisticated 3D-living constructs with spatiotemporal architecture and customized properties sensitive to external temperature and pH. Among the various external stimuli, pH- and thermo-sensitive biocompatible materials based on natural polymers are highly sought by researchers since their potential applications cover a wide range. Temperature and pH are the most affected environmental stimuli under in vitro and in vivo conditions [291]. There is interest in obtaining sensitive materials because of the excellent adhesion, biocompatibility, and biodegradability of CS. However, CS is a poor water-soluble natural polysaccharide, making it challenging to self-assemble from hydrogels.

The present contribution focuses on synthesizing and characterizing a pH- and thermo- responsive system based on CS modified with isopropyl side chains. CS-based injectable hydrogels present considerable potential for bone remodeling with sensitivity to the pH value. The use of poly (alic acid) (PAA) to modify CS is crucial for forming pH-sensitive hydrogels. These hydrogels are designed to target and control the release of drugs for treating bone diseases or infection [292,293], which can be changed by modulating the pH [117]. Lin et al. prepared PAA/CS/silica hydrogel through UV polymerization [294]. A higher compressive strength of the composite hydrogel could be achieved by forming an interpenetrating network (IPN) structure between PAA and CS with nano-silica. The hydrogel also had good biological safety. The growth factor (platelet glue) was fast and ultimately released from PAA/CS/Si hydrogel scaffolds within 620 min, which illustrated that the hydrogels are beneficial for use as scaffolds for bone defect repair. Furthermore, CS can also be modified by introducing isopropyl side-chains utilizing N-alkylation with a picoline–borane complex as a reducing agent. The altered hydrophilic/hydrophobic character endows the N-isopropyl chitosan (iCS) with the ability to form hydrogels when electrostatic chain–chain repulsion is limited. The sol–gel transition occurs at a pH compatible with the applications in the biomedical field. Considering biocompatibility, N-isopropyl chitosan (iCS) shows an interesting potential role in tissue engineering [295].

Temperature is another critical factor that influences the formation of intelligent CS hydrogels. The excellent moldable ability of thermo-responsive CS injectable hydrogels proved to be a new class of bone substitute materials that respond to temperature. This hydrogel exerts no potential adverse effects on the surrounding cells/tissues [296]. Reports have demonstrated success with thermos-sensitive CS hydrogel using poly (N-isopropylacrylamide) (PNIPAM). PNIPAM is the most studied thermo-responsive polymer, which exhibits a lower critical solution temperature (LCST) in water at around 32 °C [297]. Graft copolymerization of N-isopropylacrylamide (NIPAAm) with CS could produce a thermo-sensitive biocompatible drug-delivery carrier [298]. The hybrid hydrogel exhibits significant volume changes under the stimulation of pH and temperature. Importantly, the hydrogel system showed good cytocompatibility and rapid gelation ability without prolonged inflammatory response in vitro and in vivo. On account of the advantages of intelligent CS-based hydrogels, the synthesized dual-responsive biocompatible hydrogel has the potential to be used as an injectable hydrogel for controllable drug delivery and minimally invasive BTE applications.

Comparing intelligent CS-based hydrogels to other presently available conventional materials shows great advantages, as they have the ability of in-situ osteoblastic differentiation induction and interesting biological functions. The incorporation of growth factors and other bioactive molecules as well as nano-particles as smart modifications can enhance cell differentiation, proliferation, and attachment and therefore may improve new bone formation [299]. Even though hydrogels show instinctive superiorities in tissue engineering, many problems remain unaddressed. For instance, the interactions of injectable hydrogels with immune cells, which include macrophages, neutrophils, and dendritic cells (DCs) are still unclear and may mediate both defense and destruction. Further, the drug release mechanisms should be studied further to gain insight and thereby improve understanding of the design of more advanced and reliable hydrogel drug delivery systems.

## 9. Challenges and Future Prospects

BTE is regarded as the best alternative approach to conventional bone grafting techniques. A scaffold is an essential subunit that provides mechanical strength, a site for cell attachment, proliferation, and differentiation. Polymer selection and scaffold fabrication techniques are the key factors. The scaffold can be fabricated using various techniques based on the implantation site, the nature of polymers, and their characteristics.

CS is a readily available polymer with good biodegradability and biocompatibility. It has non-toxic, mucoadhesive, hemostatic, and antimicrobial properties employed in a broad range of biomedical and biopharmaceutical research. CS has displayed significant osteoconductivity but minimal osteoinductive properties [3]. The main limiting factor is the mechanical property, which requires strong cross-linking/blending with other materials to support bone tissue regeneration. Since CS has a pKa of 6.5 and its semi-crystalline nature favors strong intra/intermolecular hydrogen bonding, the solubility of CS at neutral pH is limited. The polycationic nature of CS can induce thrombosis, red blood cell aggregation, and hemolysis, making it unsuitable for tissue engineering applications. In addition, CS has poor antimicrobial properties at neutral pH because of the protonation of amino groups that occurs only in an acidic medium.

To solve the limitations above, several modifications at C2 or C5 positions of CS by introducing various reactive groups or by copolymerization with other polymers of interest or grafting with biological and synthetic macromolecules are explored nowadays, which render appropriate bone regenerative properties. Modified CS offers unique bio functions, such as water solubility, antibacterial properties, and pH and thermo-sensitivity. The inclusion of bioceramic materials in modified CS may facilitate the spreading of bone marrow stromal cells and significantly enhance the mechanical property [300,301,302]. These various chemical modifications can retain the original properties and expand the applications of CS in different fields such as anti-hypertensive, anti-oxidant, anti-allergy, immunology regulation, genetic material delivery, and bio-imaging [303,304]. CS modification has developed and innovated in recent years thanks to the development of ecologically friendly and biodegradable polymers and the intersection and penetration of numerous disciplines. These structural modifications are advantageous for the structure-activity relationship and could potentially contribute to the research of functional polymer materials. The value of applications in BTE would be adequately reflected with these developments of modified CS technology, while concurrently creating considerable societal and economic value.

Although investigators have fabricated a variety of biomaterials that mimic the morphology and functions of natural bone tissue and have verified the effectiveness in promoting bone regeneration at the cellular and animal levels, there are still many obstacles and challenges in the BTE application of modified CS [164,305]. Future implant designs should give an even greater consideration to local cellular responses. Adoption of this more holistic approach has the potential to increase the rate of bone tissue regeneration, providing enhanced osseointegration and improved long-term clinical success. The bio-absorption of modified CS and the regulatory mechanisms of osteocyte and osteoclast growth and development in osseointegration in vivo require further elucidation. Further, special attention should be paid to their safety, efficacy, and controllability in vivo. Not only the biomaterials but also the extensive process–property optimization should be required to achieve this goal. Demand for novel fabrication techniques will increase in the fore coming years due to their ability to design a scaffold that can be tailored for specific patient and clinical needs. It is equally valid that further fundamental and applied studies are required to improve our capacity to predict the properties of highly complex, multi-component, CS-based materials for BTE.

## 10. Conclusions

This review summarized modified CSs for their application in BTE. Modified-CS-based scaffolds/hydrogels displayed superior physical, chemical, mechanical, and biological properties, unlike their counterparts, serving as excellent vehicles for accelerating bone regeneration. Future studies should focus on optimizing novel modifying methods and evaluating their performance parameters during modifications of bone scaffolds.

This review showed that the empirical refinement of CS has candidly thrown open new avenues for the treatment of bone defects. Thus, an effort was made here to provide insights into the past and current trends in using modified CS polymers. Integration and processing of the different properties offered by various modified CS composites would be further beneficial for treating bone and bone-related defects.

## Figures and Tables

**Figure 1 ijms-23-06574-f001:**
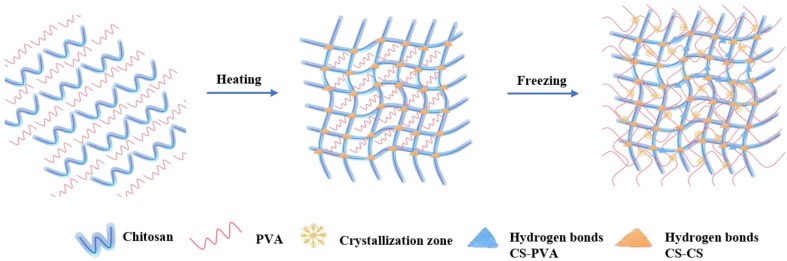
The scheme of preparing the CS/PVA hydrogel.

**Figure 2 ijms-23-06574-f002:**
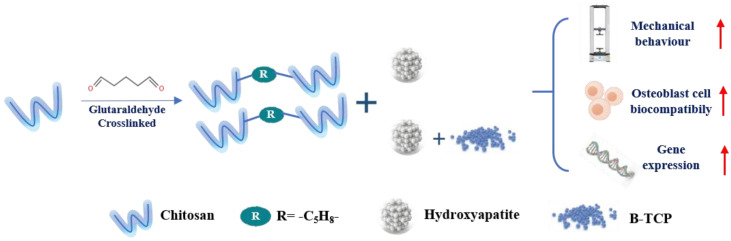
Scheme of developing and characterizing the composite scaffolds by combining biomimetically synthesized HAp nanocrystals in the presence of natural biomolecules.

**Figure 3 ijms-23-06574-f003:**
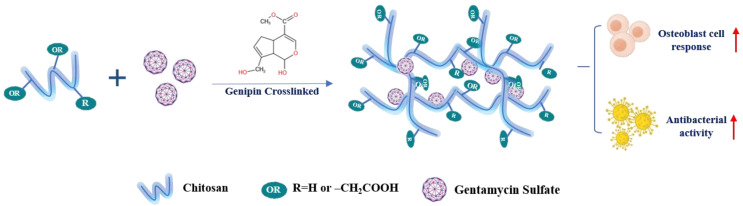
Schematic drawing of the preparation of the chitosan hydrogel.

**Figure 4 ijms-23-06574-f004:**
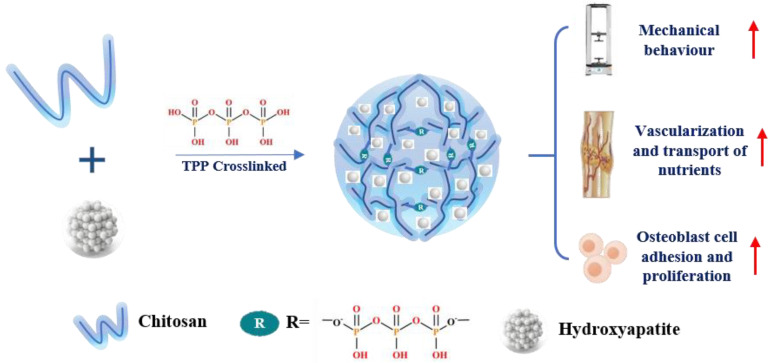
Scheme of developing and characterizing the microparticles of nano-hydroxyapatite, chitosan, and tripolyphosphate.

**Figure 5 ijms-23-06574-f005:**
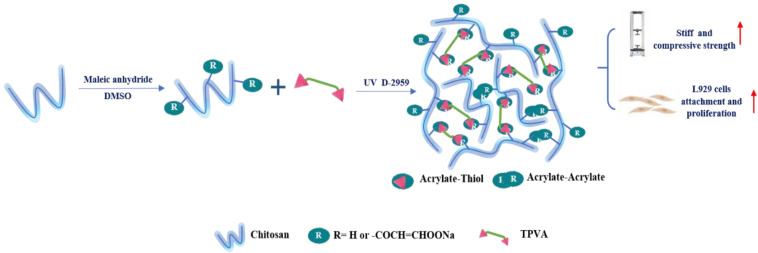
Reaction schematic and photobiology of MCS/TPVA hydrogels.

**Figure 6 ijms-23-06574-f006:**
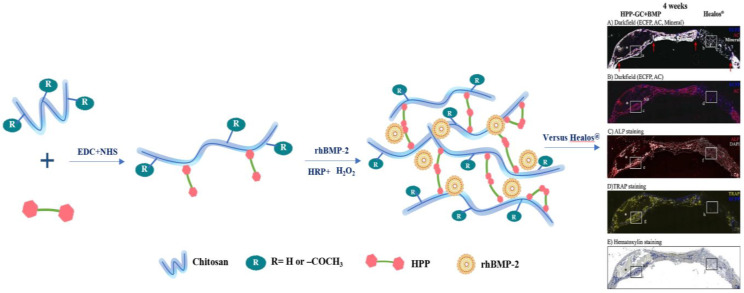
Scheme of developing and characterizing the microparticles of rhBMP-2 loaded HPP-GC hydrogel, (*) non-degraded gel, NB: new bone.

**Figure 7 ijms-23-06574-f007:**
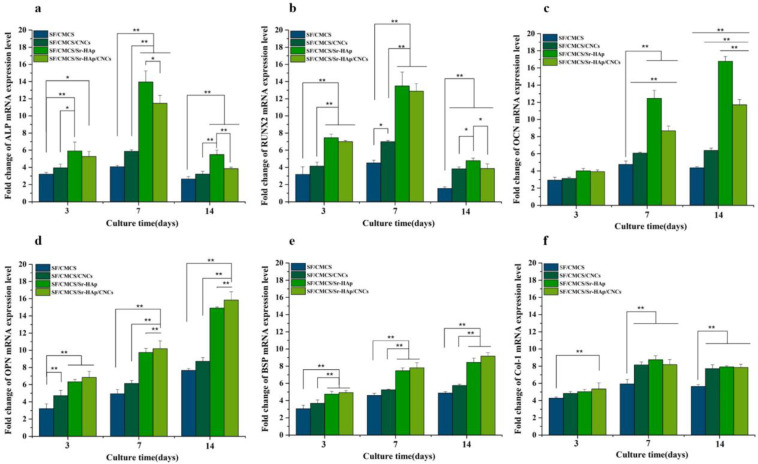
Osteogenesis gene expression of the SF-based scaffolds SF/CMCS, SF/CMCS/CNCs, SF/CMCS/Sr-Hap, and SF/CMCS/Sr-HAp/CNCs: (**a**) ALP, (**b**) RUNX2, (**c**) OCN, (**d**) OPN, (**e**) BSP, (**f**) COL-1. Probability (*p*) value of b0.05(*) is considered as significant and (*p*) value of b0.01(**) is considered to be highly significant. Copyright © [215] 2019 Elsevier B.V.

**Figure 8 ijms-23-06574-f008:**
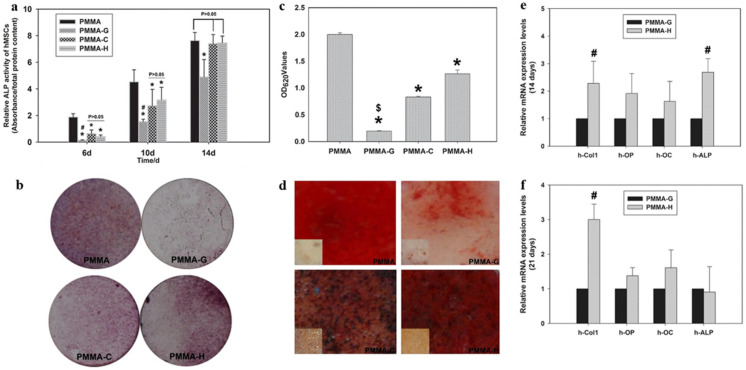
(**a**) Relative ALP activity of hMSCs after 6, 10, and 14 days of culture, (*) denotes a significantly lower ALP activity of cells than that on the PMMA surface at day 6 and day 10 (*p* < 0.01), (#) denotes a significantly lower ALP activity level compared to that on the PMMA-C and PMMA-H at day 6 and day 10 (*p* < 0.05); (**b**) Image of the positive ALP staining on the four PMMA-based bone cements on day 14; (**c**) Colorimetric quantitative analysis of the extracellular matrix mineralization on the samples after three weeks of incubation, (*) denotes significantly lower mineralization than the mineralization on the PMMA (*p* < 0.01). ($) denotes significantly lower mineralization than the mineralization on PMMA-H (*p* < 0.01); (**d**) Alizarin Red staining showing that mineralization was consistent with the quantitative analysis of mineralization; (**e**,**f**) Relative osteogenesis-related gene expressions of the hMSCs cultured on the PMMA-G and PMMA-H bone cement for 14 days and 21 days based on real-time PCR, # *p* < 0.05 compared with PMMA-G. Data were redrawn from [224].

**Figure 9 ijms-23-06574-f009:**
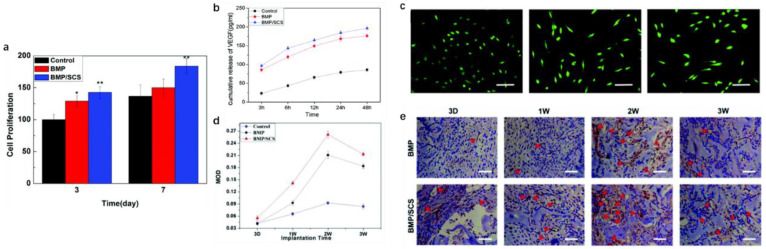
(**a**) Cell viability of BMSCs treated with BMP-2 and BMP-2/SCS determined by the MTT assay, (*) denotes significant difference (*p* < 0.05) as compared to the control group, (**) denotes *p* < 0.05 compared to both control and BMP-2 group; (**b**)Secretion kinetics of VEGF from BMSCs after culturing with BMP-2 and BMP-2/SCS. (**c**) Intracellular NO secretion of BMSCs in the culture medium (control group), BMP-2, and the BMP-2/SCS group, (**d**) Immunohistochemican anti-CD31 staining of the ectopic bone section, and (**e**) The statistical analysis of blood vessels after the microvessel counting was conducted at 200×. Probability (*p*) value of b0.05(*) is considered as significant and (*p*) value of b0.01(**) is considered to be highly significant. Data were redrawn from [230].

**Figure 10 ijms-23-06574-f010:**
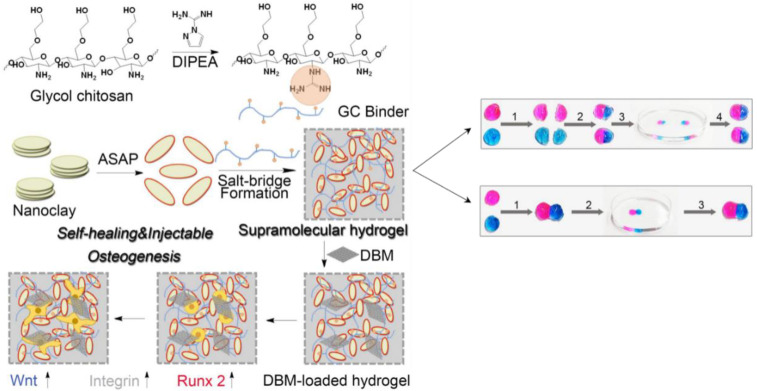
Schematic illustration of supramolecular hydrogels with self-healing and injectable properties for bone regeneration with no need for growth factors, self-healing processes of DBM-loaded supramolecular hydrogels, and of two DBM-loaded hydrogels after attachment directly. Modified from [242].

**Table 2 ijms-23-06574-t002:** Overview of CS-based biomaterials in preclinical and clinical trials.

	Animals/Volunteers (Total)	Incisions/Defects/Cells	Chitosan-Based Form	Effects	Ref.
Preclinical trials	Beagles (n/d)	Open skin wounds on the dorsal side	20 mg/wound (2 × 2 cm)	Activate immunocytes and inflammatory cells	[53]
	Mail Wistar rats (60)	Bone defects measuring 2 mm in diameter in both tibias	CS/D. ambrosioides spheres	Faster bone regeneration and a controlled release of the extract	[54]
	New Zealand white rabbits (20)	Undergoing TKA surgery and implanted with titanium rod prostheses	CMCS hydrogel	Reduce the inflammatory response around rabbit knee prostheses, affect the OPG/RANKL/RANK signaling pathway, and promote osteogenesis.	[56]
Clinical trials	Patients undergoing abdominal surgery (30)	Wound incisions	CS membrane	An effective antimicrobial and procoagulant and promote wound repair by providing a suitable environment for beneficial microbiota	[57]
	Patients aged 50–70 years old undergoing total or elective hip replacement (n/d)	Human bone marrow stromal cells	CS immobilized glasses	Stimulate fast osteoblast response, displaying rapid cell spreading and cytoskeleton reorganization	[58]

n/d: No data specified.

**Table 3 ijms-23-06574-t003:** Advantages and limitations of modified methods of chitosan.

Cross-linking Method	Strength	Limit
Physical cross-linking	◆ Safety of biomedicine Self-healing and injectable properties at room temperature	◆ Reversible ◆ Unstable Lack permanent junctions
Chemical cross-linking	◆ Easy to operate ◆ Stability ◆ Excellent mechanical properties ◆ Adjustable degradation properties Good cross-linking effect	◆ Toxic cross-linking agents ◆ Difficulty in sterilization Degree of cross-linking is not easy to control
Enzymatic cross-linking	◆ Under normal physiological conditions ◆ Highly biocompatible By modifying pH, temperature, or ionic strength, the cross-linking reaction can often be controlled	◆ Substrate specificity Most expensive crosslinkers

**Table 4 ijms-23-06574-t004:** Cross-linking mechanism of different cross-linking agents and chitosan.

Cross-Linking Agents	Cross-Linking Mechanism
Glutaraldehyde(GA)	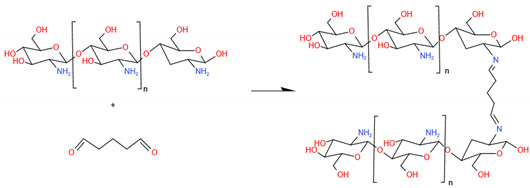
Vanillin	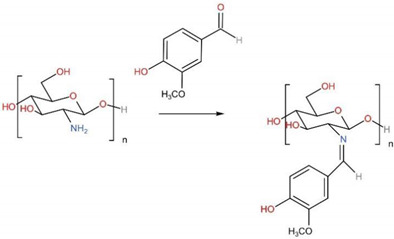
Genipin	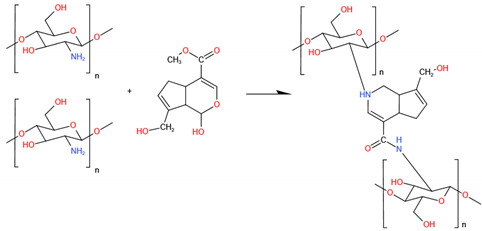
Tripolyphosphate, TPP	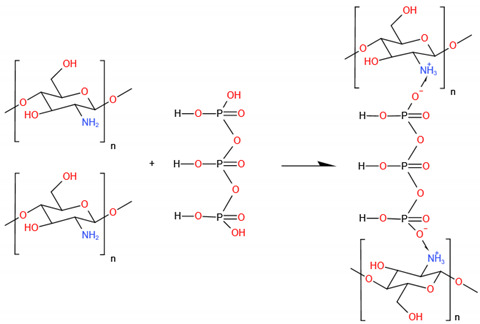
Carbodiimide(NHS/EDC)	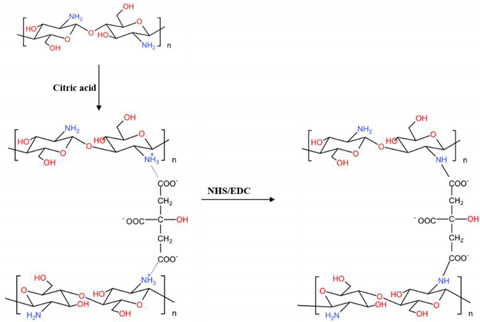
Enzymatic-cross-linking (HRP)	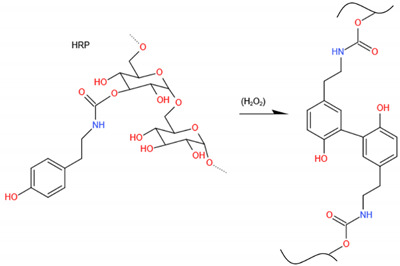
Photoinitiators	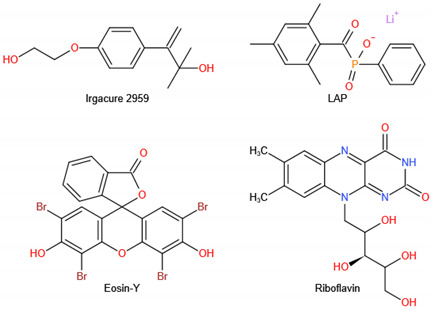
Methylmethacrylate chitosan, ChMA	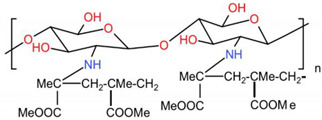

**Table 5 ijms-23-06574-t005:** Structural formula of structure-modified chitosan.

CS derivates	Chemical formula
Carboxymethyl chitosan, CMC	** 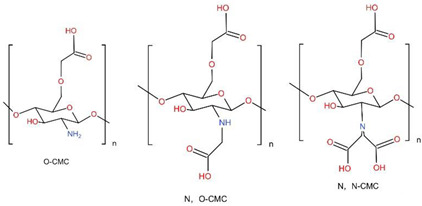 **
Hydroxypropyltrimethyl ammonium chloride chitosan, HACC	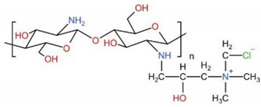
Sulfated chitosan, SCS	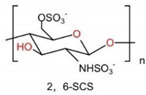
Glycol chitosan, GCS	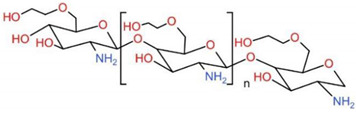
Guanidinylated chitosan, GC	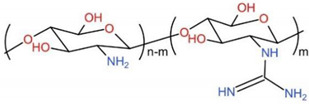

## Data Availability

Not applicable.

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
