# Peer review of "Application Progress of Modified Chitosan and Its Composite Biomaterials for Bone Tissue Engineering"

_ijms, 2022, doi:10.3390/ijms23126574_

Round 1
Reviewer 1 Report
The authors have submitted an article entitled “Application progress of modified chitosan and its composite biomaterials for bone tissue engineering" which reviews the chitosan-based biomaterials for bone tissue regeneration. The manuscript is well designed. After considering a few points, I suggest the publication of this article.
The abstract is clear and concise.
The introduction is clear and concise.
The various sections in the body text of the review are clear and concise.
The conclusions are logical.
- There are several issues regarding the English of the manuscript (grammatically). For example, in line 42 (“Recently, advances in tissue engineering and tissue engineering technology could provide a more efficient treatment”), “tissue engineering” is repeated.
- When you used first time an abbreviation, keep using the abbreviation in the rest of the text. For instance, “three-dimensional” and “3D” OR “chitosan” and “CS”.
- Authors used several subsections while have forgotten to properly assign the heading numbers. For example, in the section 3.2.1 the “crosslinking agents” must be removed, and instead other subsections get numbers like the following:
3.2.1. Aldehydic
3.2.2. Genipin
3.2.3. Tripolyphosphate (TPP) - Providing a comparison study between the effect of different crosslinking on the degradation degree and mechanical properties of the chitosan-based scaffolds is recommended.
- In the introduction part, chitosan resources, 3D printed, and 2D electrospun scaffolds should be added. Another issue is that title of this review paper is “Application progress of modified chitosan and its composite biomaterials for bone tissue engineering”. However, mostly focused on crosslinking and less focused on composites (with nanoparticles) to be reviewed. You can find more data by following these references and add to your paper. Green Chemistry, 24(1), 62-101. Carbohydrate Research, 489, 107930. Carbohydrate polymers, 227, 115364.
- A simple graphical abstract to summarize the previous works and increase the impact of this work is recommended.
Reviewer 2 Report
I suggest this paper for publication because it is well-written, timely and covers the important topic. The authors are actively working in this field. The authors did good work, however they missed an important area where chitosan is used for bone tissue engineering, that is in complex with halloysite nanoclay. The following papers are suggested for citation and expanding the work:
https://doi.org/10.1007/s42860-021-00152-7
https://doi.org/10.1016/j.carbpol.2021.118311
https://doi.org/10.1016/j.carbpol.2022.119127
These papers report the use of chitosan/halloysite in tissue engineering, including it application to bone reconstruction
Technical comments:
- figure 6 right part (microscopy) is poorly seen and needs revision.
- figure 7 - blurry, please provide good quality images
- figure 8 b and d blurry, please provide good quality images
English needs some serious revisions. There are numerous issues, e.g.:
"Recently, advances in tissue engineering and tissue engineering technology could provide a more efficient treatment" - grammar error + what is the difference between "tissue engineering" and "tissue engineering technology"? Are they not the same?
"bioproperties"?
"positive-charged"?
| Physically cross-linked could raise the stability of the chitosan through interaction |
And elsewhere in the text
Reviewer 3 Report
- The development of modified chitosan implants with improved biocompatible properties remains a significant clinical issue. Indeed, uncomplicated use of chitosan implants requires significant biochemical and biological modifications. So, assume that discussing the modification methods of chitosan scaffolds is an important scientific challenge.
- Chitosan has already been sufficiently studied, including the complications associated with its use. This may be important during the long-term follow-up period. The authors may find these articles useful [Ueno, H., Mori, T., & Fujinaga, T. (2001). Topical formulations and wound healing applications of chitosan. Advanced drug delivery reviews, 52(2), 105-115.] [Vasconcelos, D. P., Costa, M., Amaral, I. F., Barbosa, M. A., Águas, A. P., & Barbosa, J. N. (2015). Modulation of the inflammatory response to chitosan through M2 macrophage polarization using pro-resolution mediators. Biomaterials, 37, 116-123.] [Soriente, A., Fasolino, I., Gomez‐Sánchez, A., Prokhorov, E., Buonocore, G. G., Luna‐Barcenas, G., ... & Raucci, M. G. (2022). Chitosan/hydroxyapatite nanocomposite scaffolds to modulate osteogenic and inflammatory response. Journal of Biomedical Materials Research Part A, 110(2), 266-272.] [Ribeiro, J. C. V., Forte, T. C. M., Tavares, S. J. S., Andrade, F. K., Vieira, R. S., & Lima, V. (2021). The effects of the molecular weight of chitosan on the tissue inflammatory response. Journal of Biomedical Materials Research Part A, 109(12), 2556-2569.] [Renard, E., Amiaud, J., Delbos, L., Charrier, C., Montembault, A., Ducret, M., ... & Gaudin, A. (2020). Dental pulp inflammatory/immune response to a chitosan-enriched fibrin hydrogel in the pulpotomised rat incisor. Eur. Cells Mater, 40, 74-87.].
- Moreover, I suppose it is the problems of preclinical and clinical experience with chitosan that should be devoted to an essential part of the paper, maybe even separated in a special Table.
- The authors did a great job exposing the biochemical reactions, but the methods of biomodification should be uncovered as well. I mean copolymerization procedures of chitosan [Sanchez-Salvador, J. L., Balea, A., Monte, M. C., Negro, C., & Blanco, A. (2021). Chitosan grafted/cross-linked with biodegradable polymers: a review. International Journal of Biological Macromolecules, 178, 325-343.] and methods of chitosan binding with peptides, proteins, RNAs and other biomolecules [Mohammadi, Z., Eini, M., Rastegari, A., & Tehrani, M. R. (2021). Chitosan as a machine for biomolecule delivery: A review. Carbohydrate Polymers, 256, 117414.][Kulkarni, N., Shinde, S. D., Jadhav, G. S., Adsare, D. R., Rao, K., Kachhia, M., ... & Sahu, B. (2021). Peptide-Chitosan engineered scaffolds for biomedical applications. Bioconjugate Chemistry, 32(3), 448-465.].
- Table 1: Content of the Model column should be uncovered. The listing of cell types is clearly insufficient.
- Figures are collected in the special part of the paper. It is incompatible, and I propose to separate them by text.
- Some figures (#7, #8, #9) are reprints from other articles. Permissions from every journal are necessary. If there are no permissions, these figures should be removed. I recommend using the original images.
- Figure 10: the paper [152] has no sources for this figure. Which figure was modified?
- Some references are misspelled and will not be indexed. Please, check the references ##1, 15, 23, etc. For example:
9.1. Reference #1: “Jeroen: A., et al” - Jaren is a first name, not surname. The right cites will be: [Aerssens, J., Boonen, S., Lowet, G., & Dequeker, J. (1998). Interspecies differences in bone composition, density, and quality: potential implications for in vivo bone research. Endocrinology, 139(2), 663-670.]
9.2. Reference #15: The journal’s title was missed: [Kadouche, S., Farhat, M., Lounici, H., Fiallo, M., Sharrock, P., Mecherri, M., & Hadioui, M. (2017). Low cost chitosan biopolymer for environmental use made from abundant shrimp wastes. Waste and biomass valorization, 8(2), 401-406.]
9.3. All references should be double-checked to avoid misindexing.
Round 2
Reviewer 1 Report
The authors need to answer all the comments. I have not received any respond to my comments
Reviewer 3 Report
The authors have corrected most of the points, but some minor ones remain uncorrected.
1. The issue of post-implantation complications was not disclosed in the manuscript.
2. References should be at the end of a sentence or before a punctuation mark in the middle of a sentence, not after the Author's Surname et al.
3. References still need to be revised. I did not find any signs of double-checking the Reference list. For example:
Ref. #8. There is no Journal's Title, Volume and Pages of the paper: [Wu, C.; Ma, K.; Zhao, H.; Zhang, Q.; Liu, Y.; Bai, N.J.S.R. Bioactive effects of nonthermal argon-oxygen plasma on inorganic bovine bone surface.]
Ref. #9. There is no paper's Title. [L'Heureux, N., .; Germain, L., .; Auger, F.A.J.S. Tissue engineering. 1999, 284, 1621.]
Ref. #21: [Lizardi-Mendoza, J.; Monal, W.; Valencia, F. Chemical Characteristics and Functional Properties of Chitosan; Chitosan in the 981 Preservation of Agricultural Commodities: 2016.]
... and more, more... ##32, #47, ....
Round 3
Reviewer 1 Report
Comments:
There are several references published 2-decades ago like the first ref. (1998), Ref. 9. The review paper should cover the recent advances and current challenges in that specific topic. There are many reviews in those years covering that old references. It is highly recommended to use recent references (mostly last 5 years and certainly not more than 10 years ago):
2003--> 4 references //// 2004-->6 references //// 2005-->8 references //// 2006-->5 references //// 2007--->5 references
Reviewer 3 Report
Ok. In the revised form, the manuscript may be published in the the esteemed journal IJMS.
Author Response
Response to Reviewer 3 Comments (Round 3)
Thank you so much for sparing your time and efforts reviewing our paper and for the positive feedback. We are sincerely appreciated for your affirmation of our manuscript. The comments given to us about our manuscript help us for further improvements and increase the quality of our paper. Also, we have learned a lot from this process.Once again, thanks for all your comments and suggestions.